# Effectiveness of Smartwatch Guidance for High-Quality Infant Cardiopulmonary Resuscitation: A Simulation Study

**DOI:** 10.3390/medicina57030193

**Published:** 2021-02-25

**Authors:** Seong A Jeon, Hansol Chang, Sun Young Yoon, Nayeong Hwang, Kyunga Kim, Hee Yoon, Sung Yeon Hwang, Tae Gun Shin, Won Chul Cha, Taerim Kim

**Affiliations:** 1Department of Emergency Medicine, Samsung Medical Center, Sungkyunkwan University School of Medicine, Seoul 06351, Korea; sarahjun@naver.com (S.A.J.); hansol.chang@samsung.com (H.C.); wildhi.yoon@samsung.com (H.Y.); sygood.hwang@samsung.com (S.Y.H.); taegun.shin@samsung.com (T.G.S.); wc.cha@samsung.com (W.C.C.); 2Department of Digital Health, Samsung Advanced Institute for Health Sciences and Technology, Sungkyunkwan University, Seoul 06355, Korea; roseherb21@naver.com; 3Biostatistics and Clinical Epidemiology Center, Samsung Medical Center, Seoul 06351, Korea; ny.hwang@snri.co.kr (N.H.); Kyunga.j.kim@samsung.com (K.K.); 4Health Information and Strategy Center, Samsung Medical Center, 81 Irwon-ro Gangnam-gu, Seoul 06351, Korea

**Keywords:** wearable electronic devices, cardiopulmonary resuscitation, heart arrest, infant, simulation training, feedback

## Abstract

*Background and objectives:* As in adults, the survival rates and neurological outcomes after infant Cardiopulmonary resuscitation (CPR) are closely related to the quality of resuscitation. This study aimed to demonstrate that using a smartwatch as a haptic feedback device increases the quality of infant CPR performed by medical professionals. *Materials and methods:* We designed a prospective, randomized, case-crossover simulation study. The participants (n = 36) were randomly allocated to two groups: control first group and smartwatch first group. Each CPR session consisted of 2 min of chest compressions (CCs) using the two-finger technique (TFT), 2 min of rest, and 2 min of CCs using the two-thumb encircling hands technique (TTHT). *Results:* The primary outcome was the variation in the “proportion of optimal chest compression duration” and “compression rate” between the smartwatch-assisted and non-smartwatch-assisted groups. The secondary outcome was the variation in the “compression depth” between two groups. The proportion of optimal CC duration was significantly higher in the smartwatch-assisted group than in the non-smartwatch-assisted group. The absolute difference from 220 was much smaller in the smartwatch-assisted group (218.02) than in the non-smartwatch-assisted group (226.59) (*p*-Value = 0.018). *Conclusion:* This study demonstrated the haptic feedback system using a smartwatch improves the quality of infant CPR by maintaining proper speed and depth regardless of the compression method used.

## 1. Introduction

While the incidence of pediatric out-of-cardiac-arrest (OHCA) is low, that of infant OHCA is 10-fold higher, approaching the incidence in adults [1]. In addition, the survival rates for infants receiving cardiopulmonary resuscitation (CPR) are reported to be twice as poor as those for other pediatric age groups [1]. Previous studies have shown that neither the incidence nor the survival rates of pediatric OHCA have improved over the last decades [2]. As in adults, the survival rates and neurological outcomes after infant CPR are closely related to the quality of resuscitation [3,4,5]. Minimizing chest compression interruptions and allowing full chest recoil after each compression are the basic processes of the high-quality CPR [6,7].

According to the 2018 American Heart Association (AHA) guidelines for infant CPR, the compression depth should be at least one-third of the diameter of the front and back of the chest, which is approximately 1.5 inches (4 cm) [8]. The recommended compression rate is between 100 and 120 times per minute [9,10,11]. However, as stated in the guidelines, it is not easy to maintain good-quality chest compression (CC) in the field [12]. Therefore, various types of feedback devices have been developed to improve the quality of chest CCs [13,14,15].

Most of the devices are visual and auditory devices that indicated the depth, speed, and appropriateness of CCs [12]. However, the audio-visual feedback devices are difficult for users to focus on and use in crowded and noisy CPR environments [16]. In addition, these devices often need to be attached to the chest wall of patients and are designed for adults, so their use in infant CPR is limited as the CCs are carried out with two fingers [17]. Currently, wearable devices have been developed for healthcare workers, providing feedback through a smartwatch [18]. Haptic devices using vibrations from a smartwatch can be applied more effectively than audio-visual feedback devices in the field [17,19,20].

Although previous studies have shown that a smartwatch with real-time feedback can improve CPR quality, most were limited to adult CPR [21]. Furthermore, previous studies on infant CPR have focused on haptic feedback and medical students [17]. This study aimed to determine the effectiveness of smartwatch feedback in the delivery of high-quality infant CPR by medical professionals.

## 2. Materials and Methods

### 2.1. Study Design

This was a prospective, randomized, case-crossover simulation study that aimed to demonstrate the effect of a smartwatch-type haptic device on the quality of CC in infant CPR. We conducted a randomized, controlled mannequin-simulation study of a single rescuer hands-only CPR at the ED of Samsung Medical Center from 1 to 31 December 2019. The study was approved by the institutional review board of Samsung Medical Center (No. 2019-07-042-007) on 2 December 2019. All procedures were carried out in accordance with the Declaration of Helsinki.

### 2.2. Study Participants

Based on previous studies on adult CPR with haptic devices, we calculated our sample size. A size of 36 participants per group was calculated using the McNemar test for two paired proportions, with 0.025 significance and 0.80 power. The proportion of discordant pairs was assumed to be 50%; based on that, the proportion of optimal duration of CCs using a smartwatch was 95% and that of CCs without using a smartwatch was 45%. We hypothesized that there is no difference in the optimal duration between the two-finger technique (TFT) and the two-thumb encircling hands technique (TTHT).

Participants were recruited via recruitment information on the employee notice board. Medical professionals such as doctors, nurses, and emergency medical technicians (EMTs) who were either Pediatric Advanced Life Support (PALS), Basic Life Support (BLS), or Advanced Cardiovascular Life Support (ACLS) certified or had previous experience with infant CPR were eligible for participation. Volunteers were excluded if they had cardiovascular or musculoskeletal disease. Informed consent was attained from all the participants.

### 2.3. Study Protocol

All study participants were provided an introduction to the study protocol and the optimal cardiac compression for CPR suggested by the AHA guidelines that lasted for 10 min. At the first visit, the participants were randomly allocated to two groups: control first group (Group A, *n* = 18) and the smartwatch first group (Group B, *n* = 18). Each session consisted of 2 min of CCs using TFT, 2 min of rest, and 2 min of CCs using the TTHT. All participants performed hands-only CPR according to the AHA guideline.

Group A performed two sets of CC (2 min each) without the feedback of the smartwatch at the first visit. CC sets comprised 2 min of TFT and 2 min of TTHT. The smartwatch first group (Group B) performed CCs while wearing a smartwatch with the preinstalled app at the first visit. The participants were then crossed over to the other group and performed CCs after a washout period of more than 6 h. The study process is illustrated in Figure 1. A rate of 100–120 CCs/min was considered the optimal rate. The total number of CCs was measured during the 2 min, and 0.5 to 0.6 s per one CC was considered the optimal duration.

### 2.4. Description of the Devices

“Laerdal^®^ Resusci baby” was used to gather and store the CC performance data in a laptop. Chest compressions are performed on the ‘Resusci baby’ mannequin and the number and depth of chest compressions performed by the subject are set to be recorded in the program linked to the mannequin. The “Samsung^®^ Galaxy Gear S3” smartwatch (Samsung Electronics Inc., Seoul, Republic of Korea) with a haptic metronome application (on the Galaxy Store app called Wearable Metronome) was used in this study. This smartwatch was to be worn around the wrist and was set to provide constant vibrations at the rate of 110/min (Figure 2).

### 2.5. Measures

We gathered demographic information such as age, sex, job, and infant CPR experience from each subject. For outcome measures, the number of CCs during 2 min, chest compression duration, and compression depth per stroke were collected. Thirty-second intervals were defined as quartiles in a sequence: first quartile (1Q), second quartile (2Q), third quartile (3Q), and fourth quartile (4Q).

### 2.6. Outcomes

The primary outcome was the variation in the “proportion of optimal chest compression duration” and “compression rate”. We compared the compression rate between the two groups. The secondary outcome was the variation in the “compression depth” between two groups. Subgroup analyses were further conducted to investigate the effect of smartwatch guidance for each compression technique, i.e., TFT and TTHT, and for every 30-s quartile.

### 2.7. Data Analysis

The demographic characteristics of each group were analyzed. Continuous variables with normal distribution were presented as mean (standard deviation (SD)) and those with non-normal distribution, as medians with interquartile rages (IQRs). Categorical variables were presented as counts and percentages. Two-sample *t*-test was used to analyze the data of two continuous variables with normal distribution, and Mann-Whitney U test was used to analyze the data of variables that were not normally distributed. Chi-square test was used to compare the frequencies of categorical variables between two groups.

We used the generalized estimation equation (GEE) models and odds ratio (ORs) with 95% confidence intervals to analyze repeatedly measured variables during multiple time points. In the GEE models, the measured variables were the CC duration and depth. The use of smartwatch and the type of compression technique were included as explanatory variables, and the interaction effect was also analyzed along with their primary effects. Subgroup analyses were further carried out for each technique. Statistical significance was set at 95% level (*p-*Value of <0.05). Statistical analysis was performed using Statistical Analysis System (SAS) version 9.4 (SAS Institute, Cary, NC, USA) and R 3.5.1 (R Foundation for Statistical Computing).

## 3. Results

In this study, 36 participants were recruited and randomly allocated to two groups. Of them, 28 were physicians, 6 were paramedics, and 2 were nurses. None of the participants dropped out. Table 1 shows the demographic characteristics of the study population. The smartwatch first group showed less experience of CC within 1 month (smartwatch first group: 2.0 (0.0–5.0) vs. 4.0 (3.0–5.0), *p* = 0.032); otherwise, there were no significant differences between the two groups in terms of age, sex, job, work experience, and experience of CC within 5 years.

### 3.1. Result for CC Performance

The CC performance results are shown in Table 2. The proportion of optimal CC duration was significantly different between the smartwatch-assisted CCs and non-smartwatch-assisted CCs. Both groups achieved satisfactory mean CC duration of 0.5–0.6 s (smartwatch-assisted CCs (550.8 ± 54.8 msec) vs. non-smartwatch-assisted CCs (529.9 ± 68.4 msec), *p-*Value < 0.001). However, the proportion of optimal CC duration was higher in the smartwatch-assisted CCs (smartwatch-assisted CCs (11,081/71.1%) vs. non-smartwatch-assisted CCs (9,630/59.4%), *p-*Value < 0.001).

The results of subgroup analysis for compression duration between the groups are shown in Table 3. The average duration of smartwatch-assisted CCs was about 18 ms longer than that of non-smartwatch-assisted CCs, with statistical significance (*p* < 0.001). The estimates for 2Q, 3Q, and 4Q were, respectively, 9.39 (95% CI, 4.69–14.09), 12.95 (6.78–19.13), and 13.69 (7.44–19.93) seconds longer than the estimate for Q1, with statistical significance (*p* < 0.001). We conducted interaction analyses between with or without smartwatch-assisted CCs and four interquartiles. The results showed that the effect of smartwatch-assisted CCs was insignificantly different along the four interquartiles. Smartwatch-assisted CCs showed generally increasing quality from 1Q to 4Q. In another subgroup analysis between TFT and TTHT, the CC duration did not show a significant difference.

We compared the compression rate between the smartwatch-assisted CCs and non-smartwatch-assisted CCs. The total number of CCs was counted for 2 min. We set the average value of 110/min as the adequate speed for the smartwatch metronome and 220 CCs in 2 min as the appropriate rate. Table 4 shows that the compression rate of the smartwatch-assisted group was 8.56 times lower than that of the non-smartwatch-assisted group, with statistical significance (*p* = 0.018). In the subgroup analysis between TFT and TTHT, the compression rate did not show a significant difference. The absolute difference from 220 was much smaller for the smartwatch assisted CCs (1.98) than for non-smartwatch-assisted CCs (6.59). The results are shown in Table 5.

### 3.2. Results for Compression Depth

The mean CC depth was not significantly different between the smartwatch-assisted and non-smartwatch-assisted groups. The subgroup analysis showed that the adjusted odds ratios (AORs) for 2Q, 3Q, and 4Q in comparison with 1Q for the compression depth were 0.58 (95% CI, 0.43–0.79), 0.49 (0.34–0.72), and 0.40 (0.27–0.59), respectively (Table 6). The interaction analysis shows the AOR of the smartwatch-assisted CCs and its positive impact. The AORs for 2Q, 3Q, and 4Q in comparison with 1Q were 1.49 (95% CI, 1.03–2.14), 1.80 (1.13–2.87), and 2.16 (1.33–3.52), respectively (Table 6). The appropriateness of the compression depth had gradually increased from 1Q to 4Q in smartwatch-assisted CCs (Figure 3).

In another subgroup analysis, the TTHT group achieved the adequate compression depth 8.5 times better than the TFT group (*p-*Value < 0.001). The AOR was 8.49 (95% CI, 4.45–16.19) for the TTHT in comparison with the TFT. The TTHT produced stronger and more consistent CCs than the TFT without the compromising the quality of compressions.

## 4. Discussion

This study showed that the haptic feedback system using a smartwatch improves the quality of infant CPR by maintaining proper speed and depth regardless of the compression methods. The proportion of optimal CC duration was higher and the absolute difference from the adequate compression rate was much smaller in the smartwatch-assisted group than in the non-smartwatch-assisted group. The participants were able to maintain the CCs at a constant duration over time when assisted by a smartwatch. Although the mean CC depth was not significantly different between the groups, subgroup analysis showed that the CC depth of the smartwatch-assisted group was appropriate over time.

The prompt feedback on the duration of CCs may result in an add-on effect of enhancing the CCs [22]. Previous studies have shown that inappropriate compression rates may lead to poor quality of compressions [23]. During cardiac compression, if the compression rate is misjudged, it may be difficult to maintain continuous CC rates, eventually leading to a decrease in compression depth [22]. The survival rate and neurological outcomes after infant CPR are closely related to the quality of resuscitation [2]. Hence, smartwatches can be used by medical professionals to enhance the efficiency of infant CPR.

Recently, the development of wearable electronic devices is attracting attention given the advances in health care. Furthermore, these devices can help overcome the differences in medical resource quality between low-income and high-income countries [24]. Previously, audio-visual feedback devices have been used during CPR, which were not effective in noisy environments, had spatial constraints, and were expensive [12,14,16,25]. However, now we can improve the quality of infant CPR with a handy smartwatch that is not limited by spatial constraints, is cost-effective, and is easy to use. Smartwatch feedback provides generally consistent CC duration and depth without being affected by the compression method.

Previous studies have been often been limited to adult CPR [21,26] because it is difficult to apply existing devices to infants and the incidence of infant CPR is low. The other factor is that in infant CPR, there are two types of CC methods involving the use of two fingers, whereas in adult CPR, both hands are used [12]. It has been reported that real-time feedback can improve the quality of CC even with increased fatigue during infant CPR [15]. We proved that smartwatch feedback provides generally consistent CC duration and depth during compression statistically. Therefore, the haptic feedback system using a smartwatch may help maintain high-quality CC despite increased fatigue.

While previous studies focused on medical students, our study involved medical professionals [17]. In fact, medical professionals engage more often in adult CPR than in infant CPR given its low incidence, so these haptic feedback devices could be helpful in clinical settings [1]. We can expect the smartwatch feedback to be effective when the educational effect declines [27].

We examined the effects of the use of smartwatch on the two controversial methods (TFT vs. TTHT) in our subgroup analysis. Previous studies have reported that chest compression using two fingers increases fatigue and reduces compression quality [28]. Between TFT and TTHT, the optimal CC duration and rate did not show a significant difference. However, the TTHT group achieved adequate compression depth about 8.5 times better than the TFT group (*p-*Value < 0.001). Thus, the TTHT provided stronger and more consistent CCs than the TFT without compromising the quality of compression.

This study has several limitations. First, this was a simulation study that used a mannequin, so the results have limited applicability to real clinical settings. We assumed that haptic feedback using vibrations from a smartwatch can be effective in noisy environments. However, in real clinical practice, there may be factors that cause interference with the compressors’ performance in crowded environments [12]. Research with haptic devices at actual medical sites should be considered in future. Second, the intensity of the vibration from the smartwatch cannot be modified and varies according to producers. Participants may feel vibration intensity differently due to individual perceptions and CC motion [21]. If the vibration intensity is too weak or strong, it can affect the quality of compression. Therefore, further studies are required to optimize the intensity of vibration via smartwatches for individual users. Third, because the actual resuscitation situation is often noisy and urgent, it can affect the time to wear the smartwatch and operate the application, and these aspects were not assessed in this current study. Further studies are needed to identify factors that affect the time to operate the feedback system. Fourth, it would be less uncomfortable to wear a smartwatch during infant CPR due to two fingers being used, whereas both hands are used in adult CPR. However, depending on the individual, it may be uncomfortable to wear a smartwatch and perform chest compressions. Using a flexible and wearable form of haptic device or thin, soft rubber-band smartwatch can be an alternative to the inconvenience of wearing a hard smartwatch during CPR [29]. Therefore, further studies are required to assess how comfortable it is to individuals performing CPR with a smartwatch.

## 5. Conclusions

In conclusion, using a smartwatch as a feedback device ensures the delivery of high-quality CPR to infants by medical professionals. In this simulation study, we showed that smartwatch feedback improves the adequacy of the CC duration and depth regardless of the compression method used.

## Figures and Tables

**Figure 1 medicina-57-00193-f001:**
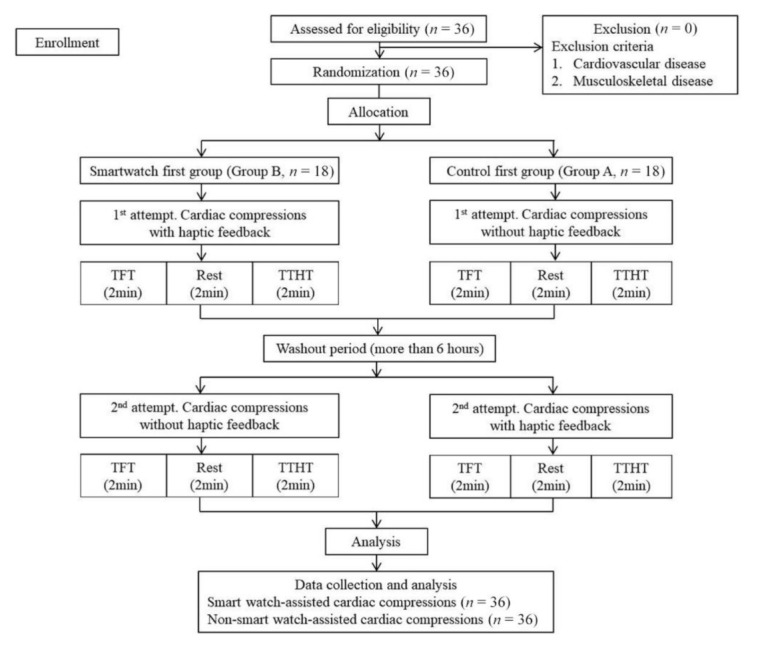
CONSORT flow diagram of the study. TFT, two-finger technique; TTHT, two-thumb encircling hands technique; min, minute.

**Figure 2 medicina-57-00193-f002:**
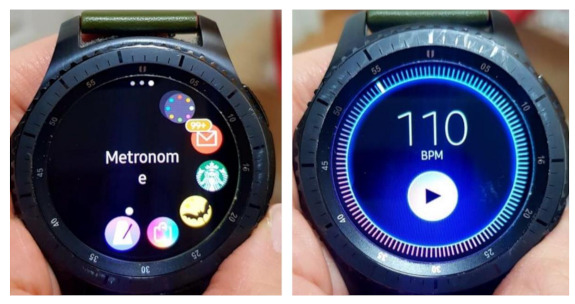
The smartwatch application is set to 110 BPM. BPM, beats per minute.

**Figure 3 medicina-57-00193-f003:**
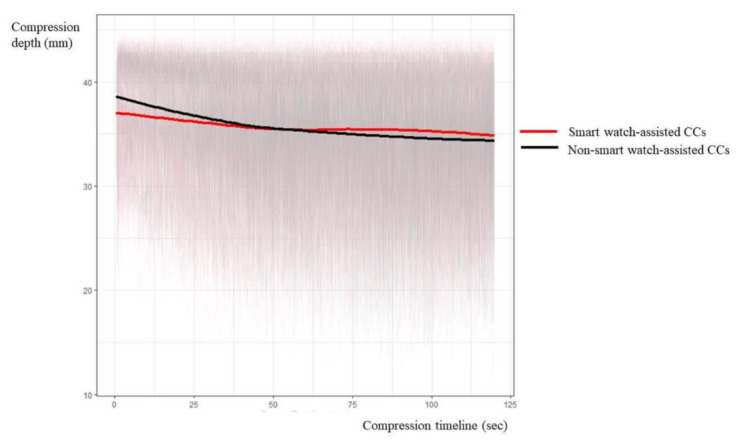
Comparison of the timeline graphs of compression depth between smart watch-assisted CCs and non-smartwatch-assisted CCs. CC, chest compression.

**Table 1 medicina-57-00193-t001:** Baseline characteristics between smartwatch first group and control first group.

	Smartwatch First Group (*n* = 18)	Control First Group (*n* = 18)	*p-*Value
Age, median (IQR)	28.5 (25.0–30.0)	27.5 (25.0–31.0)	0.622
Sex, *n* (%)			1
Male	6 (33.3%)	7 (38.9%)	
Female	12 (66.7%)	11 (61.1%)	
Job, *n* (%)			0.298
Doctor	12 (66.6%)	16 (88.8%)	
EMT	5 (27.8%)	1 (5.6%)	
Nurse	1 (5.6%)	1 (5.6%)	
Work experience, *n* (%)			1
Up to 3 years	9 (50.0%)	8 (44.4%)	
More than 3 years	9 (50.0%)	10 (55.6%)	
Experience of cardiac compression within 5 years (IQR)	30.0 (10.0–50.0)	30.0 (20.0–50.0)	0.662
Experience of cardiac compression within 1 month (IQR)	2.0 (0.0–5.0)	4.0 (3.0–5.0]	0.032

Values are presented as median with IQR or number (proportion).

**Table 2 medicina-57-00193-t002:** Proportion of optimal CC duration between smartwatch-assisted and non-smartwatch-assisted groups.

	Smartwatch -Assisted CCs	Non-Smartwatch -Assisted CCs	*p-*Value
Total number of chest compression	15,585	16,212	
Proportion of optimal chest compression (%)	11,081 (71.1%)	9630 (59.4%)	<0.001
Chest compression duration (msec)	550.8 ± 54.8	529.9 ± 68.4	<0.001

CC, chest compression; msec, millisecond. The values are presented as mean ± standard deviation or number (%).

**Table 3 medicina-57-00193-t003:** Subgroup analysis for compression duration between smartwatch-assisted and non-smartwatch-assisted groups.

Parameter	Level (vs. Ref.)		Estimate	Standard Error	95% Confidence Limits		Z	*p-*Value
Group	Smartwatch-assisted CCs vs. Non-smartwatch-assisted CCs		18.15	4.79	8.75	27.54	3.78	<0.001
Quartile	2Q vs. 1Q		9.39	2.40	4.69	14.09	3.92	<0.001
	3Q vs. 1Q		12.95	3.15	6.78	19.13	4.11	
	4Q vs. 1Q		13.69	3.19	7.44	19.93	4.3	
Method	TTHT vs. TFT		−0.32	4.53	−9.19	8.55	−0.07	0.943
Group* quartile	Smartwatch-assisted CCs vs. Non-smartwatch-assisted CCs	2Q vs. 1Q	0.86	2.40	−3.85	5.57	0.36	0.266
		3Q vs. 1Q	3.99	3.50	−2.86	10.84	1.14	
		4Q vs. 1Q	6.16	3.44	−0.58	12.89	1.79	

TFT, two-finger technique; TTHT, two-thumb encircling hands technique; Q, quartile; 1Q, first quartile; 2Q, second quartile; 3Q, third quartile; 4Q, Fourth quartile.

**Table 4 medicina-57-00193-t004:** Comparison of compression rate between smartwatch-assisted and non-smartwatch-assisted groups and TFT vs. TTHT group.

Parameter	Level (vs. Ref.)	Estimate	Standard Error	95% Confidence Limits		Z	*p-*Value
Group	Smartwatch-assisted CCs vs. Non-smartwatch-assisted CCs	−8.56	3.63	−15.68	−1.45	−2.36	0.018
Method	TTHT vs. TFT	4.03	2.11	−0.11	8.16	1.91	0.057

CC, chest compression; TFT, two-finger technique; TTHT, two-thumb encircling hands technique.

**Table 5 medicina-57-00193-t005:** Absolute difference in compression rate from set value (220 times).

Average Compressions	Smartwatch -Assisted CCs	Non-Smartwatch -Assisted CCs	Overall
TFT	216.01	224.57	220.29
TTHT	220.04	228.60	224.32
Overall	218.02	226.59	222.31
Absolute difference from 220	1.98	6.59	

CC, chest compression; TFT, two-finger technique; TTHT, two-thumb encircling hands technique. Values are presented as mean.

**Table 6 medicina-57-00193-t006:** Subgroup analysis for compression depth between smartwatch-assisted and non-smartwatch-assisted groups.

Parameter	Level (vs. Ref.)		Estimate	Standard Error	95% Confidence Limits		Z	*p−*Value	Odds Ratio	95% CI of OR	
Group	Smartwatch-assisted CCs vs. Non-smartwatch assisted CCs		−0.44	0.27	−0.96	0.08	−1.67	0.096	0.64	0.38	1.08
Quartile	2Q vs. 1Q		−0.54	0.16	−0.85	−0.24	−3.48	<0.001	0.58	0.43	0.79
	3Q vs. 1Q		−0.71	0.19	−1.09	−0.34	−3.71		0.49	0.34	0.72
	4Q vs. 1Q		−0.91	0.20	−1.30	−0.52	−4.56		0.40	0.27	0.59
Method	TTHT vs. TFT		2.14	0.33	1.49	2.78	6.49	<0.001	8.49	4.45	16.19
Group* Quartile	Smartwatch-assisted CCs vs. Non-smartwatch-assisted CCs	2Q vs. 1Q	0.39	0.19	0.03	0.76	2.13	0.015	1.49	1.03	2.14
		3Q vs. 1Q	0.59	0.24	0.12	1.05	2.47		1.80	1.13	2.87
		4Q vs. 1Q	0.77	0.25	0.28	1.26	3.11		2.16	1.33	3.52

TFT, two-finger technique; TTHT, two-thumb encircling hands technique; Q, quartile; 1Q, first quartile; 2Q, second quartile; 3Q, third quartile; 4Q, fourth quartile.

## Data Availability

The data for this study has been archived at the Department of Emergency Medicine, Samsung Medical Center, Sungkyunkwan University School of Medicine, Seoul, Korea. The data is private property of Samsung Medical Center, Sungkyunkwan University School of Medicine. The data is available on request as per the University Policy.

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
