# Peer review of "Effectiveness of Smartwatch Guidance for High-Quality Infant Cardiopulmonary Resuscitation: A Simulation Study"

_medicina, 2021, doi:10.3390/medicina57030193_

Round 1

Reviewer 1 Report

This study aimed to evaluate if using a smartwatch as a haptic feedback device increases the quality of infant CPR performed by medical professionals.

The study is interesting and provides information on how the quality of infant CPR can be improved. Nonetheless, there are some issues that may be better addressed.

The abbreviations should be defined at their first appearance in the text (example “EMTs who were either PALS, BLS, or ACLS” lines 88-89), even in the abstract (CPR line 21).

“While the incidence of out-of-cardiac-arrest (OHCA) is low, that of infant OHCA is 41 10-fold higher, approaching the incidence in adults...  ” (lines 41-42). It is not clear what the authors wanted to say here

“t is not easy to maintain” (line 54) maybe the authors wanted to say “it is not easy to maintain”

In material and methods, the authors should give some information about the model of the mannequin used for simulations.

“We compared the compression rate between the two groups” (line 193) – should be rephrased.

The authors identified very well some of the limitations of the study. Since this study is about infant CPR, wearing a smartwatch is not uncomfortable during CPR, but from my point of view wearing this kind of device during adult CPR is very uncomfortable as each time when I have to perform this maneuver I have to take off my watch.  My suggestion for the future studies of the authors would be to evaluate also how comfortable are the CPR performers to wear smartwatches when they perform cardiac massage.

Reviewer 2 Report

Overall, this is a good job. The introduction is quite complete and the proposed objective is clear and concise. Regarding the methodology used, the fact of using a randomized study to select the participants makes the results much more explicit. The number of study participants may be too low, especially considering that those chosen have been healthcare professionals
The fact of comparing the two techniques used to perform CPR on infants with Laerdal manikins should have meant taking much more data than just the number of compressions and their mean depth, despite having distributed the quartiles over time  
